# Posterior Reversible Encephalopathy Syndrome in Children with Malignancies or After Hematopoietic Cell Transplantation: A Polish Nationwide Study

**DOI:** 10.3390/cancers17233789

**Published:** 2025-11-26

**Authors:** Tomasz Brzeski, Wanda Badowska, Katarzyna Mycko, Patrycja Tyszka, Martyna Korzeniewicz, Julia Kolodrubiec, Wojciech Mlynarski, Karolina Gawle-Krawczyk, Katarzyna Koch, Pawel Laguna, Aleksandra Kiermasz, Agnieszka Mizia-Malarz, Marta Malczewska, Katarzyna Drabko, Anna Malecka, Ninela Irga-Jaworska, Patrycja Marciniak-Stepak, Katarzyna Derwich, Jacek Wachowiak, Magdalena Bartnik, Tomasz Ociepa, Tomasz Urasinski, Malgorzata Sawicka-Zukowska, Maryna Krawczuk-Rybak, Grzegorz Waliszczak, Walentyna Balwierz, Szymon Skoczen, Tomasz Jarmolinski, Krzysztof Kalwak, Iwona Ruranska, Tomasz Szczepanski, Wioletta Bal, Radosław Chaber, Magdalena Tarasinska, Bozenna Dembowska-Baginska, Agnieszka Chodala-Grzywacz, Grazyna Karolczyk, Sonia Pajak, Monika Richert-Przygonska, Krzysztof Czyzewski, Jan Styczynski

**Affiliations:** 1Department of Pediatric Oncology and Hematology, University of Warmia and Mazury in Olsztyn, Regional Specialized Children’s Hospital, 10-561 Olsztyn, Poland; 2Department of Pediatrics, Oncology and Hematology, Medical University of Lodz, 92-213 Lodz, Poland; 3Department of Pediatric Oncology and Hematology, Clinical Transplantology and Pediatrics, Medical University of Warsaw, 02-091 Warsaw, Poland; 4Department of Pediatric Hematology and Oncology, Medical University of Silesia, 40-752 Katowice, Poland; 5Department of Pediatric Hematology, Oncology and Transplantology, Medical University of Lublin, 20-093 Lublin, Poland; 6Department of Pediatrics, Hematology and Oncology, Medical University of Gdansk, 80-952 Gdansk, Poland; 7Department of Pediatric Oncology, Hematology and Transplantology, Poznan University of Medical Sciences, 60-572 Poznan, Poland; 8Department of Pediatrics and Hemato-Oncology USK1, Pomeranian Medical University, 71-252 Szczecin, Poland; 9Department of Pediatric Oncology and Hematology, Medical University of Bialystok, 15-274 Bialystok, Poland; 10Department of Pediatric Oncology and Hematology, University Children’s Hospital of Krakow, 30-663 Krakow, Poland; 11Department of Pediatric Oncology and Hematology, Institute of Pediatrics, Jagiellonian University Medical College, 30-663 Krakow, Poland; 12Department of Pediatric Hematology, Oncology and BMT, Wroclaw Medical University, 20-529 Wrocław, Poland; 13Department of Pediatric Hematology and Oncology, Medical University of Silesia, 41-800 Zabrze, Poland; 14Department of Pediatrics, Faculty of Medicine, University of Rzeszow, 35-310 Rzeszow, Poland; 15Clinic of Pediatric Oncology and Hematology, State Hospital 2, 35-301 Rzeszow, Poland; 16Department of Oncology, Children’s Memorial Health Institute, 04-730 Warsaw, Poland; 17Department of Pediatric Hematology and Oncology, Regional Polyclinic Hospital in Kielce, 25-734 Kielce, Poland; 18Division of Pediatric Hematology and Oncology, Chorzow City Hospital, 41-500 Chorzow, Poland; 19Department of Pediatric Hematology and Oncology, Collegium Medicum in Bydgoszcz, Nicolaus Copernicus University in Torun, 85-094 Bydgoszcz, Polandjstyczynski@cm.umk.pl (J.S.)

**Keywords:** PRES, children, pediatric oncology, HCT, predictive index, risk factors

## Abstract

Posterior reversible encephalopathy syndrome (PRES) is a significant complication of oncological and transplant treatment in children. We present the results of a multicenter, nationwide study focusing on PRES in children treated for malignancies or undergoing hematopoietic cell transplantation. Clinical and laboratory abnormalities, management strategies, outcomes, and complications of PRES were analyzed. The occurrence of PRES had a negative impact on survival. We paid special attention to the prodromal period of PRES. In our opinion, this may be the key to its early diagnosis. We propose the predictive index, as well as diagnostic criteria and a new name for the syndrome, saving the existing acronym: we have shown that the disease can no longer be regarded as “reversible” and we propose the use of the word “rapid” in the PRES acronym.

## 1. Introduction

PRES is a constellation of specific clinical symptoms accompanied by characteristic neuroimaging findings. PRES often complicates the treatment of the underlying disease, delays therapy, and frequently requires hospitalization in the intensive care unit (ICU) [1,2,3]. It occurs both in adults and children undergoing treatment for malignancies [1,4,5], hematological disorders [6,7], kidney diseases [8,9], obstetric complications [10,11], and infections [12,13] after hematopoietic cell or solid organ transplants [8,14,15], as well as in cases of immunodeficiency or immunosuppression [16,17] and autoimmune diseases [18,19]. The name PRES was first introduced in 1996 by Judith Hinchey [20]. The pathogenesis of PRES is not yet fully understood. The leading hypothesis assumes that factors contributing to the development of PRES include impaired autoregulation of cerebral circulation caused by endothelial injury. Many studies reported the impact of particular drugs on PRES. More than seventy substances have been identified as potentially associated with this syndrome [21]. The drugs considered most often include the following: calcineurin inhibitors, vincristine, antifungal agents, bevacizumab, and rituximab; however, PRES episodes connected with novel drugs such as TKI and anti-CD19 CAR-T were also reported [5,21,22,23,24,25].

The main clinical symptoms of PRES include seizures, altered consciousness, hypertension, headaches, visual disturbances, and vomiting [1,4,5,26,27,28,29,30]. Their severity is individual, ranging from mild to severe with life-threatening episodes. There are differences in the prevalence of individual symptoms between children and adults. For example, abdominal pain and constipation are common in pediatric patients, but are not considered significant issues in adults [1,4,5].

Neuroimaging is essential for the diagnosis of PRES, with MRI being the preferred method [1,28]. It typically reveals the vasogenic brain edema. The edema is usually asymmetric but in most cases, it is bilateral. It always involves the subcortical white matter and often the gray matter of the cerebral cortex as well [1,2,31,32]. Bartynski et al. distinguished three main radiological patterns of PRES, which account for approximately 70% of all cases [1,2,31].

There are no standardized treatment protocols of PRES. Management is symptomatic and usually includes general rescue measures, anticonvulsant therapy, hypotensive drugs, and possible elimination of suspected causative factors [1,28,30,33,34,35].

The objective of this study was the analysis of clinical and laboratory abnormalities, management strategies, outcomes, and complications of PRES in a large cohort of children treated for malignancies or undergoing HCT.

## 2. Materials and Methods

### 2.1. Design of the Study

In this nationwide, multicenter, retrospective study, pediatric patients treated for malignancy at pediatric hematology and oncology (PHO) centers or undergoing HCT between 2014 and 2022, who were diagnosed with PRES, were analyzed. Follow-up was completed on 31 December 2022. The study was approved by the local Bioethical Committee at the University of Warmia and Mazury in Olsztyn, Poland. Patients were assigned to either the study group or the control group.

### 2.2. Inclusion Criteria

To be eligible for inclusion in the study group, patients had to meet all of the following three criteria: diagnosis of malignancy between the ages 0 and 18, a PRES episode diagnosed between 2014 and 2022, and a PRES episode that occurred during anticancer treatment or following HCT. To be eligible for the control group, patients had to meet all of the following three criteria: diagnosis of malignancy between the ages 0 and 18, no history of a PRES episode, and undergoing anticancer treatment or HCT between 2014 and 2022. PRES was diagnosed locally based on clinical and radiological symptoms and signs. Each case was verified by two independent researchers during the data collection phase. For each patient with PRES, 2–3 controls were selected locally. All HCT patients in both groups had a diagnosis of malignancy. Every patient diagnosis of PRES was made in a local pediatric oncology center by both an oncologist and a radiologist, based on clinical symptoms and cranial MRI. Prior to the final decision to include the patient in the study, the diagnosis was confirmed by two researchers coordinating multicenter PRES project (TB and JS) and a second independent radiologist.

### 2.3. Data Analyzed

In both groups, data on malignancy or transplantation, past medical history, and family history were collected. In the study group, detailed information related to PRES episodes including symptoms, laboratory abnormalities, treatment, and complications of PRES was also recorded reported.

### 2.4. Definitions

PRES was diagnosed in patients with characteristic neuroimaging findings (vasogenic edema) accompanied by a combination of specific clinical symptoms such as consciousness disturbances, seizures, hypertension, apathy, headache, and visual disturbances [1,2,3].

Central nervous system (CNS) involvement at the time of initial malignancy diagnosis was defined as the presence of clinical signs (e.g., cranial nerve palsy), imaging findings (e.g., CNS infiltration or mass), or cerebrospinal fluid (CSF) abnormalities (WBC > 5/µL, positive cytospin for blasts) in cases of hematological malignancies, the presence of a primary CNS tumor, or CNS metastases in cases of solid tumors.

Status epilepticus was defined as a seizure lasting 30 min or more, or a series of seizures occurring without recovery of consciousness between episodes within a 30 min period.

### 2.5. Endpoints of Analysis

The primary endpoint of the study was overall survival (OS). The secondary endpoints included disease-free survival (DFS), ICU hospitalization, and chronic complications. Overall survival (OS) was calculated from the date of cancer diagnosis to the date of death from any cause, last follow-up, or study completion, and disease-free survival (DFS) was calculated from the date of diagnosis to the date of recurrence or death (whichever occurred first), last follow-up, or study completion.

### 2.6. Statistical Analysis

Numerical traits were presented as mean and standard deviation (SD) or median and interquartile range (IQR), depending on their distribution. Categorical traits were described with n and % of group. Normality was verified with Shapiro–Wilk test, skewness, and kurtosis. Levene’s test was used to assess variance homogeneity. Comparisons were performed with t-Student test, t-Welch test, Mann–Whitney U test, Pearson’s chi-square test, or Fisher’s exact test, as appropriate. Additional Benjamini–Hochberg adjustment of *p*-values, due to multiple testing, was used for comparisons of the group with PRES and the control group, and its outcomes were presented in attachment. Logistic regression analysis was run in 2 steps. First, univariate models were run for all predictors. Next, predictors with *p*-values not higher than 0.250 were selected for the multivariate model. Final variables selection for the multivariate model was based on a stepwise procedure [36]. Model verification was based on the Negelkerke R2 and Hosmer–Lemeshow GOF test. ROC analysis was performed to determine the predictive capacity of the factors with the highest importance in the multivariate analysis. The Kaplan–Meier method was used to estimate survival, along with 95% CI (5-year OS and 5-year DFS). Survival differences between study groups were verified with log rank test. *p*-value < 0.05 was assumed as significant. Statistical analysis was performed in R software (version 4.1.2).

## 3. Results

### 3.1. Demographics

A total of 438 children from seventeen PHO centers and HCT units were included in the study, comprising 120 patients in the study group and 318 in the control group. Among the patients diagnosed with PRES, 93 of 120 children were treated for de novo disease, 17 for relapse, and 10 had undergone HCT. The most common type of malignancy was ALL, diagnosed in 92 patients (76.7%). CNS involvement was identified in 19 of these ALL cases. PRES was diagnosed at a mean age of 8.7 ± 4.0 years. The median time from de novo diagnosis to onset PRES was 1.5 months, and 1.6 months in relapse cases. In patients who underwent HCT, the median time from transplantation to onset PRES was 4.7 months. PRES episodes occurred significantly earlier in patients with de novo or relapsing disease than in those who had undergone HCT (*p* = 0.029), see Figure 1.

### 3.2. Comparison of Study Group (PRES) vs. Control Group (Non-PRES)

Patients in the PRES group were older than those in the non-PRES group; the mean age in the PRES group was 8.0 years vs. 6.9 years in the non-PRES group (*p* = 0.019). PRES patients were more likely to have a diagnosis of hypertension (*p* = 0.039) or epilepsy (*p* = 0.015) before malignancy. PRES patients were more likely to have hypertension (*p* < 0.001), seizures (*p* < 0.001), and hospitalization in the ICU (*p* < 0.001) during treatment. The PRES group had a lower remission rate at the end of the follow-up (*p* < 0.001) and a higher mortality rate (*p* < 0.001) (Table 1). Results additionally adjusted for multiple testing are presented in the attachment (Appendix A). After this adjustment, the previously significant differences for hypertension before malignancy and hypertension during follow-up became non-significant, although they remained at the level of a statistical trend.

### 3.3. Procedures Preceded PRES Within 7 Days

Within 7 days before PRES: 37/120 (30.8%) patients underwent a lumbar puncture, 67/120 (55.8%) received red blood cell transfusion, and 54/120 (45%) received platelet transfusion. Overall, 77/120 (64.2%) patients received antifungal treatment, including posaconazole 32/120 (26.7%), micafungin 26/120 (21.7%), caspofungin 7/120 (5.8%), voriconazole 7/120 (5.8%), ABLC 2/120 (1.7%), and L-AMB 2/120 (1.7%).

### 3.4. Clinical Symptoms of PRES

The most common clinical symptoms present at the onset of the PRES episode included consciousness disturbances (84.2%; n = 101), hypertension (74.2%; n = 89), and apathy/fatigue (64.2%; n = 77). Seizures occurred in 96 out of 120 children (80%), of which 62 (64.6%) were generalized tonic–clonic, 14 (15.0%) were focal, and 9 (9.4%) were focal seizures with secondary generalization. Another seizure type was reported in 11 (11.5%) children. Seizures, disturbances of consciousness, hypertension, and visual disturbances occurred most frequently on the day of PRES episode diagnosis. However, some symptoms preceded the onset of PRES by a median of 2–5 days, with abdominal pain, limb pain, and apathy being among the earliest manifestations (Figure 2a,b).

### 3.5. Laboratory Abnormalities in PRES

The most common laboratory abnormalities were as follows: electrolyte disturbances (75%; n = 90; including hyponatremia in n = 59 and hypokalemia in n = 45 patients), elevated CRP concentration (51.7%; n = 62) and hypertransaminasemia (44.2%; n = 53). PRES episodes were preceded by a variety of laboratory abnormalities developing over a median time of 2–5 days, with elevated CRP concentration occurring earliest (Figure 3a,b). On the day of PRES, the median hemoglobin concentration (Hgb) was 10.00 g/dL (range 6.6–15.3 g/dL), platelet count (PLT) was 90 × 109/L (range 7–556 × 109/L), and white blood cell count (WBC) was 1.6 × 109/L (range 0–21.00 × 109/L). Appendix A presents the types of disease in specific laboratory abnormalities observed during the prodromal period.

### 3.6. Treatment of PRES

Treatment of PRES included antihypertensive management, therapy of seizures, and use of steroids and mannitol. The most commonly used antihypertensive drugs included the following: amlodipine (41.7%; n = 50), furosemide (36.7%; n = 44), captopril (27.5%; n = 33), enalapril (16.7%; n = 20), urapidil (10.8%; n = 13), metoprolol (9.2%; n = 11), labetalol (8.3%; n = 10), propranolol (6.7%; n = 8), spironolactone (6.7%; n = 8), clonidine (5.0%; n = 6), and nifendipine (5.0%; n = 6). The most commonly used anticonvulsant drugs were as follows: diazepam (69.2%; n = 83), clonazepam (30.8%; n = 37), phenobarbital (23.3%; n = 28), valproic acid (18.3%; n = 22), levetiracetam (17.5%; n = 21), midazolam (8.3%; n = 10), and pregabalin (0.8%; n = 1). General anesthetics drugs were required for 15 (12.5%) children. Steroids were administered during PRES in 44 (36.7%) children, 10.8% of whom received them as part of the treatment protocol. Dexamethasone was used in 22 (18.3%) patients, hydrocortisone in 4 (3.3%) patients, and methylprednisolone in 2 (17%) patients. Mannitol was used in 39.2% of the children (n = 47).

### 3.7. PRES Complications

The early complications of PRES led to hospitalization in the ICU, which was necessary for 50% of the children in the study group (n = 60). For 35.8% (n = 43) of patients, admission to the ICU was caused by PRES whereas for 14.2% (n = 17) of children, it was caused by another reason. The most common cause of hospitalization in the ICU due to PRES was status epilepticus, which occurred in 26 children (21.7%). Late complications of PRES were observed in 41.7% of patients (n = 50), mainly hypertension (22.5%; n = 27) or epilepsy (20.8%; n = 25). Neuropsychiatric complications were diagnosed in six children (5.0%).

### 3.8. PRES in Children After HCT (N = 10)

Allo-HCT was performed before the PRES episode on 9/120 patients (7.5%; ALL relapse 5/9, AML relapse 4/9) whereas auto-HCT was performed on 1/120 children. Five children had their first HCT, four children had their second HCT, and one patient had their third HCT. In the control group, HCT was performed on 36/318 children (11.3% of the group). Patients with PRES had conditioning more often with fludarabine (6/10 vs. 5/36, *p* = 0.001) or melphalan (4/10 vs. 1/36, *p* = 0.002) and less frequent using TBI (0/10 vs. 18/36; *p* = 0.006) or etoposide (0/10 vs. 17/36, *p* = 0.003). In the HCT-PRES group, seizures (*p* < 0.001) were observed more frequently. Among the patients in the study group, remission at the end of follow-up was less common (4/10 vs. 30/36; *p* = 0.012) and the mortality rate was higher (6/10 vs. 8/36, *p* = 0.047). There were no differences in the type of transplant, source of HSCs, type of donor, HLA compatibility, and frequency of aGvHD, cGvHD, or VOD.

### 3.9. Risk Factors Analysis of PRES Development

In the univariate analysis, the following risk factors were significant for development of PRES: hypertension before malignancy (OR = 4.57; 95%CI [1.10; 22.54]; *p* = 0.040), epilepsy before malignancy (OR = 5.53; 95%CI [1.43; 26.53], *p* = 0.017), age at diagnosis (continuous variable, per 1 year, OR = 1.06; 95%CI [1.01; 1.11]; *p* = 0.019), treatment due to relapse (OR = 2.02; 95%CI [1.03; 3.89]; *p* = 0.037), treatment for malignancies other than ALL or AML (OR = 1.98; 95%CI [1.11; 3.48]; *p* = 0.019), occurrence of hypertension during therapy (OR = 17.42; 95%CI [10.52; 29.59]; *p* < 0.001) and occurrence of seizures during therapy (OR = 56.57; 95%CI [30.84; 108.91]; *p* < 0.001) (Table 2). In the multivariate analysis, two factors had an impact on the occurrence of PRES: hypertension during therapy (OR = 23.60; 95%CI [10.36; 61.86]; *p* < 0.001) and seizures during therapy (OR = 86.58; 95%CI [36.81; 237.86]; *p* < 0.001) (Table 2).

### 3.10. Determination of the Predictive Capacity of Selected Factors for PRES

Seizures and hypertension during treatment had high predictive value for PRES as characterized by a high area under the curve (AUC). For seizures, the AUC was 0.867, the sensitivity was 80%, the specificity was 93%, the PPV was 0.82, and the NPV was 0.93 (*p* < 0.001). For hypertension, the AUC was 0.800, the sensitivity was 74%, the specificity was 86%, the PPV was 0.60, and the NPV was 0.90 (*p* < 0.001) (Figure 4).

### 3.11. PRES Predictive Index

To identify patients at high risk of developing full-symptomatic PRES (with seizures and consciousness disturbances), we developed a PRES predictive index. The index incorporates the six most frequent clinical symptoms and laboratory findings that preceded PRES onset. The highest score is assigned to hypertension, a well-established risk factor for the PRES, characterized by high sensitivity and specificity. Specific electrolyte disturbances, which occurred in 75% of patients before PRES diagnosis, represent the most objective criterion. These abnormalities typically preceded PRES by a median of 2 days for patients with hyponatremia and 4 days for those with hypocalcemia. Apathy was a symptom observed in the majority of patients and preceded PRES with a median range of 4 days. While less common, abdominal pain, headaches, or visual disturbances were often components of the PRES clinical presentation and tended to appear before seizures. A score ≥ 5 points was considered a positive result (Table 3). In the study group, a positive PRES predictive index score was identified in 103 out of 120 patients (85.8%).

### 3.12. Survival Analysis

At the end of follow-up, remission was observed in 92/120 children (76.7%), 2/120 patients were undergoing intensive treatment, and 1/120 patients was a loss of observations. Relapse occurred in 13/120 children (10.8%) after PRES. The median time from PRES diagnosis to relapse was 22.9 months (range 10.9–56.2 months). The mortality rate in the PRES group was 25/120 (20.8%). Deaths were not directly connected to the PRES episode. The median time from PRES diagnosis to death was 14.4 months (range 0.5–62.3).

Overall survival (OS) at 5 years in the PRES group was 82.0% vs. 93.3% in the control group (*p* = 0.006); 5-year disease-free survival (DFS) in the PRES group was 62.4% vs. 85.3% in the control group (*p* < 0.001) (Figure 5a,b). The median follow-up for the study group was 38.5 months vs. 34.3 months for the control group.

Overall, 60/120 (50%) patients in the PRES group required treatment in the ICU (PRES-ICU subgroup). OS at 5 years in the PRES-ICU group was 73.3% vs. 90.6% in the PRES-non-ICU group (*p* = 0.029); DFS at 5 years in the PRES-ICU group was 58.4% vs. 66.3% in the PRES-non-ICU group (*p* = 0.144) (Figure 5c,d).

## 4. Discussion

PRES is a significant clinical issue in pediatric oncology and transplantology. Assuming an average number of 1150 children are diagnosed for malignancy each year in Poland, including 230 children with ALL, we estimated the frequency of PRES occurring in Poland for one episode per every 85 children with malignancy (1.2%) and one episode per every 30 children with ALL (3.0%) [37].

The main findings of our analysis of this large cohort of children who developed PRES during chemotherapy or after HCT include the identification of risk factors, prodromal clinical symptoms, and laboratory abnormalities, early and late complications of PRES, and the impact of PRES occurrence on survival.

### 4.1. Risk Factors

Our results indicate that the risk of PRES may increase with age. Anastasopoulou et al. reported that the median age of children with PRES was significantly higher than that of those without PRES (8.5 vs. 4.4 years), OR 1.1/year [4]. Similarly, Thavamani et al. reported an association between older age and PRES (OR 1.02; *p* < 0.001) [8]. Other results were given by Sommers et al., who reported no effect of age on PRES development, although the PRES group was small (14 patients) [27]. Impact of older age on PRES remains unclear. Further studies are needed.

The majority of PRES episodes occurred in children with hematological malignancies. Similar trends were reported by Cordeli et al. with 86.3% of patients affected [26], and by Zama et al. who reported that 72.3% of pediatric PRES cases occurred in patients with leukemia or lymphoma [28]. We did not confirm that the T-ALL phenotype or CNS involvement predisposes PRES, which contrasts with the findings of Anastasopoulou et al. and Banerjee et al. [4,5], but it is consistent with the results reported by Zama et al. [28].

Some reports indicate that specific drugs may increase the risk of PRES development. Banerjee et al. suggested a potential association between more frequent vincristine dosing and an increased risk of PRES, emphasizing that concomitant use of azoles and VCR may further increase this risk [5]. Moreover, multiple studies have linked the development of PRES to the use of calcineurin inhibitors [6,21,24,25].

### 4.2. Clinical Symptoms and Laboratory Abnormalities

Majority of PRES episodes in ALL occurred during induction chemotherapy. In the study by Banerjee et al., 28/29 children with PRES were diagnosed during protocol I [5]. A similar conclusion was reported in a study by Zama et al., where 65/67 PRES cases occurred during the first 30 days of ALL chemotherapy; these authors emphasize that PRES occurred later in children undergoing HCT than after chemotherapy [28]. Our results also correspond with those presented by Anastasopoulou et al. where 28/52 PRES episodes were observed during induction therapy [4]. Seizures, consciousness disturbances, and hypertension are the most common symptoms of PRES. However, the absence of seizures or normal blood pressure does not exclude the diagnosis of PRES [4,5,6,26,27,28,30]. We also observed that gastrointestinal symptoms are common in children with PRES. In the study by Anastasopoulou et al., abdominal pain affected 53.8% of children, constipation in 51.9%, and nausea in 20% of patients [4]. Similarly, Banerjee et al. found that abdominal pain and constipation occurred in 72% and 79% of children, respectively [5].

Electrolyte disturbances are frequently observed in the clinical course of PRES. Anastasopoulou et al. reported hyponatremia in 70.5% of children with PRES, hypocalcemia in 41.9%, and hypo- or hypermagnesemia in 25.6% of patients [4]. Similarly, Banerjee et al. reported hyponatremia in 79.3% and hypomagnesemia in 6.9% of children [5]. In contrast, Hun et al. identified elevated levels of sodium, potassium, and magnesium as protective factors against the development of PRES [3].

### 4.3. Early and Chronic Complications

PRES increases the risk of ICU hospitalization. In the study by Sommers et al., 79% of children with PRES required ICU admission, and chemotherapy had to be delayed in 36% of the children [27]. In the study by Anastasopoulou et al., ICU admissions due to PRES were reported in 64.7% of patients [4]. Similarly, Cordelli et al. found that 42.3% of patients with PRES required ICU care [26]. PRES is a known risk factor for the development of epilepsy and neurological deficits. In the publication by Cordelli et al., 6.3% of the study participants were diagnosed with epilepsy following PRES, and 1.8% developed neurological deficits [26]. Anastasopoulou et al. reported that 13.5% of children were diagnosed with epilepsy after PRES, with the same percentage developing neurological deficits [4].

### 4.4. Survival

The pathophysiology of reduced survival after PRES in children undergoing anticancer therapy or HCT is not fully understood, but it has been documented in both original studies and a systematic review. One of the possible hypotheses assumes that delaying therapy due to severe general condition during PRES may reduce treatment efficacy and increase the risk of progression or relapse; however, there are no prospective studies confirming this [5]. In the study by Sommers et al., the mortality rate among patients with PRES was 29% [27], while Cordelli et al. reported a 33.3% mortality rate, including one death directly attributed to PRES [26]. Similarly Thavamani et al. reported a mortality rate of 3.2% in the PRES group, compared with 0.4% in the non-PRES group (*p* < 0.001) [8]. Unlike other researchers, Anastasopoulou et al. reported that despite a significant percentage of children with PRES requiring hospitalization in the ICU (64.7%), mortality was 3.8%, and deaths were not directly related to PRES [4]. This result aligns with the findings of Fugate et al. [1], who reported a mortality rate of 3–6% among adult PRES patients with various diseases. In pediatric patients with ALL, Banerjee et al. reported that PRES was associated with significantly reduced OS (*p* = 0.040) and EFS (*p*= 0.001) [5]. Similarly, a systemic review by Hun et al., involving 418 children with PRES, reported an overall mortality rate of 21%; however, in most cases, deaths were related to underlying disease progression or treatment-related toxicity, rather than to PRES itself [3]. Perhaps without well-designed further prospective study focusing on the impact of PRES on survival, this issue may remain in the area of uncertainty.

### 4.5. Diagnostic Criteria and Predictive Index

We proposed diagnostic criteria for PRES (Table 4). Establishing clear and widely accepted diagnostic criteria may improve communication among researchers and facilitate multicenter and international studies.

Previously published studies on PRES provided data on the type and frequency of symptoms. Until now, no report emphasized a prodromal period. We included data on clinical symptoms and laboratory abnormalities. Further research is needed to confirm their significance in the pathogenesis and diagnosis of PRES. Based on our observations, we proposed the predictive index (Table 3), which will be validated in a prospective study. We assume that a positive result (≥5 points) should qualify the patient for CNS MRI. This approach may enable earlier detection of PRES and open new avenues for intervention strategies, such as the prophylactic use of anticonvulsants in the pre-seizure period, which may help prevent status epilepticus, the most common reason for ICU admission. Another important direction for future research is identifying the most effective and safe antihypertensive management strategies. Considering the galloping progress of machine learning processes, the use of AI-assisted tools in PRES diagnostics is an interesting future research area [38].

### 4.6. Summary of Discussion

Given the high risk of chronic complications and reduced long-term outcomes, the term “reversible” in PRES no longer accurately reflects the clinical course of the condition. We propose renaming the entity to “posterior rapid encephalopathy syndrome”, which better aligns with the pathophysiology while preserving the widely recognized acronym PRES. Furthermore, we advocate for the development of evidence-based recommendations for the diagnosis, treatment, and long-term management of PRES.

The main limitations of this study are its retrospective design and the subjective selection of patients for the control group. However, the study has several notable strengths. To our knowledge, this is the largest cohort of pediatric patients with PRES associated with malignancies or with HCT reported to date. Additional strengths include its multicenter design and the extensive volume of clinical and laboratory data collected. The novel findings presented in this study contribute valuable insights into PRES in the pediatric population. A particularly innovative aspect is the emphasis placed on the prodromal period preceding PRES onset, as well as the detailed characterization of its clinical course and the therapeutic interventions applied.

## 5. Conclusions

PRES is a significant complication of oncological and transplant treatment in children. It occurs most frequently during initial chemotherapy or in the early post-HCT period. The majority of affected patients were children with ALL. Seizures and hypertension were the most common clinical symptoms of PRES; whereas electrolyte disturbances were the most frequent laboratory abnormalities. Admission to the ICU was significantly more common in the PRES group; with status epilepticus being the leading cause. Epilepsy and persistent hypertension were the most frequent long-term complications of PRES. The occurrence of PRES had a negative impact on survival. Based on our findings, we proposed diagnostic criteria and developed a predictive index for PRES. Given the numerous chronic complications observed, the term “reversible” may be misleading. Therefore, we propose reconsidering the terminology used to describe this condition. In addition, we emphasize the urgent need for standardized recommendations for the diagnosis and management of PRES in pediatric patients.

## Figures and Tables

**Figure 1 cancers-17-03789-f001:**
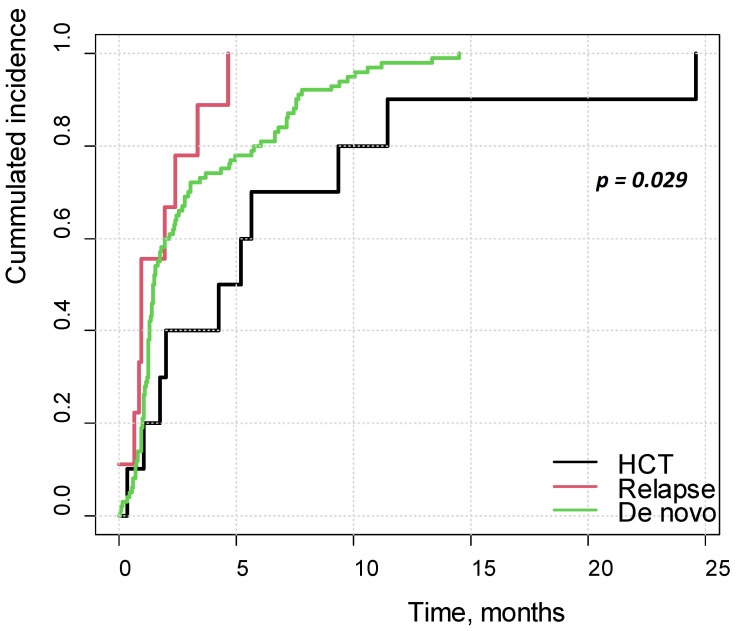
Cumulative incidence of PRES in patients with de novo disease, with relapse and after HCT.

**Figure 2 cancers-17-03789-f002:**
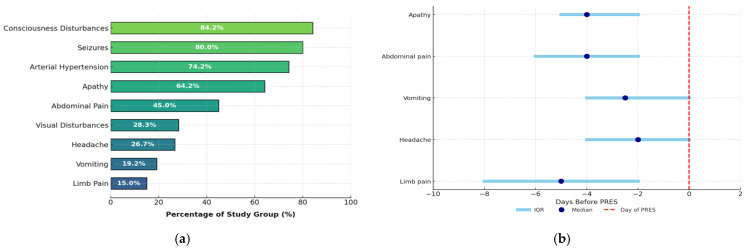
Clinical symptoms of PRES: (**a**) frequency and (**b**) time period preceding PRES.

**Figure 3 cancers-17-03789-f003:**
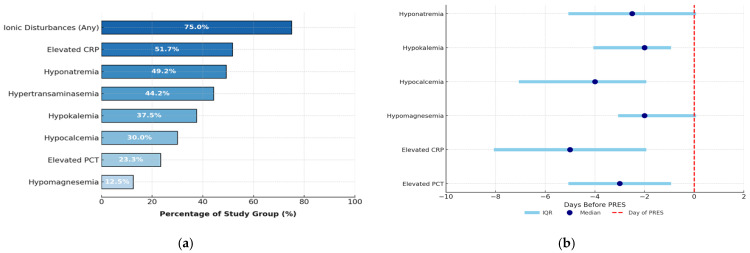
Laboratory abnormalities in PRES: (**a**) frequency and (**b**) time period preceding PRES.

**Figure 4 cancers-17-03789-f004:**
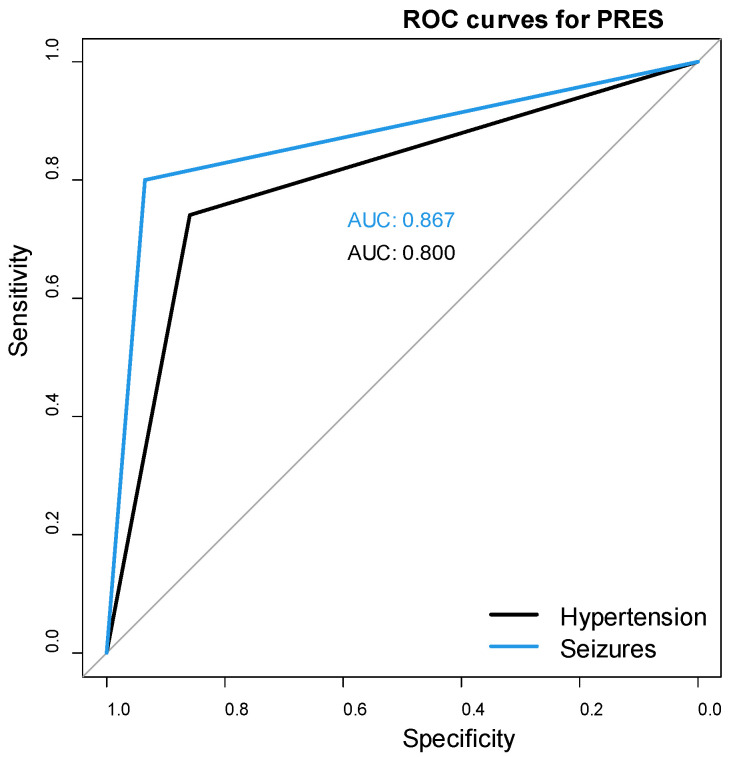
ROC curves for hypertension and seizures as diagnostic factors for PRES.

**Figure 5 cancers-17-03789-f005:**
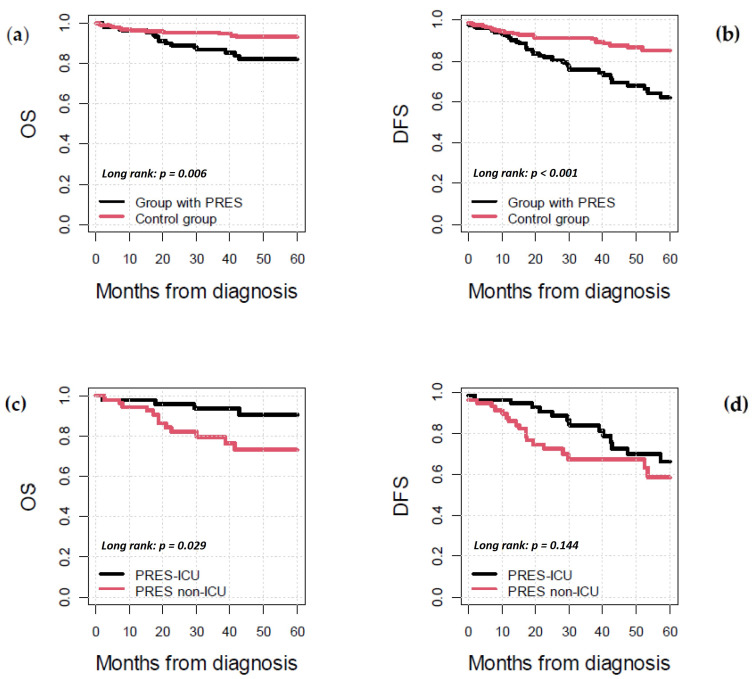
Survival curves, months from malignancy diagnosis: (**a**) 5-year OS study group vs. control group; (**b**) 5-year DFS study group vs. control group; (**c**) 5-year OS PRES-ICU vs. PRES non-ICU group; and (**d**) 5-year DFS PRES-ICU vs. PRES non-ICU group.

**Table 1 cancers-17-03789-t001:** Characteristics of study group vs. control group.

Parameter	Study Group (PRES)	Control Group (Non-PRES)	*p*
Number of patients (%)	120 (100.0)	318 (100.0)	-
Female [n (%)]	44 (36.7)	137 (43.1)	0.268
Male [n (%)]	76 (63.3)	181 (56.9)	0.268
Week of gestation 37–42 * [n (%)]	65 (90.3)	186 (89.9)	>0.999
Apgar < 8 pts. * [n (%)]	1 (1.4)	4 (2.0)	>0.999
Epilepsy before malignancy [n (%)]	6 (5.0)	3 (0.9)	0.015
Hypertension before malignancy [n (%)]	5 (4.2)	3 (0.9)	0.039
Hypertension in the family [n (%)]	5 (4.2)	10 (3.1)	0.567
Developmental delay [n (%)]	6 (5.0)	18 (5.7)	0.972
Age at diagnosis, years, [M ± SD]	7.99 ± 3.77	6.89 ± 4.54	0.019
Treatment for relapse	17 (14.2)	24 (7.5)	0.053
Age at relapse, years, [M ± SD]	10.11 ± 4.82	10.56 ± 4.55	0.808
HCT [n (%)]	10 (8.3)	36 (11.3)	0.462
Age at HCT, years, [M ± SD]	12.58 ± 4.68	10.66 ± 4.56	0.281
Disease			0.057
Acute lymphoblastic leukemia [n (%)]	92 (76.7)	273 (85.8)	
Acute myeloid leukemia	4 (3.3)	9 (2.8)	
Other malignancies [n (%)]	24 (20.0)	36 (11.3)	
CNS involvement [n (%)]	23 (19.2)	53 (16.7)	0.635
Hypertension during treatment [n (%)]	89 (74.2)	45 (14.2)	<0.001
ICU admission [n (%)]	60 (50.0)	94 (29.6)	<0.001
Seizures during treatment [n (%)]	96 (80.0)	21 (6.6)	<0.001
Hypertension during follow-up	27 (22.5%)	45 (14.2)	0.035
Epilepsy during follow-up	25 (20.8%)	0 **	<0.001
Remission at the end of follow-up [n (%)]	92 (76.7)	298 (93.7)	<0.001
Death [n (%)]	25 (20.8)	21 (6.6)	<0.001

* The percentage is given in relation to all patients with information on the week of gestation (study group: n = 72, control group: n = 207) and Apgar score (study group: n = 71, control group: n = 204). ** Data available for 189 patients from the control group.

**Table 2 cancers-17-03789-t002:** Univariate and multivariate analysis for PRES development.

Parameter	Univariate Analysis	Multivariate Analysis
OR	95% CI	*p*	OR	95% CI	*p*
Sex						
Female	1	-	-	1	-	-
Male	1.31	0.85–2.03	0.225	-	-	-
Age at diagnosis, years	1.06	1.01–1.11	**0.019**	-	-	ns
Relapse treatment	2.02	1.03–3.89	**0.037**	-	-	ns
Age at diagnosis of relapse, years	0.98	0.82–1.16	0.799	-	-	-
HCT	0.71	0.33–1.43	0.365	-	-	-
Disease						
ALL	1	-	-	-	-	-
AML	1.32	0.35–4.15	0.652	-	-	-
Other neoplasms	1.98	1.11–3.48	**0.019**	-	-	ns
CNS involvement	1.19	0.68–2.02	0.538	-	-	-
Hypertension during therapy *	17.42	10.52–29.59	**<0.001**	23.60	10.36–61.86	**<0.001**
Hypertension before malignancy	4.57	1.10–22.54	**0.040**	-	-	ns
Hypertension in the family	1.34	0.41–3.85	0.601	-	-	-
Seizures during therapy *	56.57	30.84–108.91	**<0.001**	86.58	36.81–237.86	**<0.001**
Epilepsy before cancer	5.53	1.43–26.53	**0.017**	-	-	ns
Seizure before cancer	2.34	0.74–7.18	0.134	-	-	-
Week of gestation 37–42	1.05	0.44–2.76	0.918	-	-	-
Apgar < 8 points	0.71	0.04–4.93	0.765	-	-	-
Developmental delay	0.88	0.31–2.15	0.787	-	-	-

* In all patients from the study group, hypertension and seizures occurred in prodromal period before diagnosis of PRES.ns—non statistically significant; bold—statistically significant parameter.

**Table 3 cancers-17-03789-t003:** PRES predictive index.

Criteria	Points
Hypertension with sudden onset	4
Specific electrolyte disturbances *	3
Apathy/fatigue	2
Abdominal pain	1
Headache	1
Visual disturbances	1

* Hyponatremia < 131 mmol/L or hypokalemia < 3.1 mmol/L or hypomagnesemia < 0.66 mmol/L or hypokalcemia < 2.05 mmol/L.

**Table 4 cancers-17-03789-t004:** Diagnostic criteria of PRES.

Clinical Criteria (at Least One)	Radiological Criterium (Mandatory)
1. Seizures or impaired consciousness	Characteristic findings in MRI: vasogenic edema of the subcortical white matter ± gray matter of the cerebral cortex
2. Hypertension	
3. Apathy/fatigue	
4. Headache	
5. Abdominal pain	
6. Visual disturbances	

## Data Availability

The data supporting the conclusions of this article will be made available by the authors on reasonable request.

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
