# Peer review of "Posterior Reversible Encephalopathy Syndrome in Children with Malignancies or After Hematopoietic Cell Transplantation: A Polish Nationwide Study"

_cancers, 2025, doi:10.3390/cancers17233789_

Round 1
Reviewer 1 Report (New Reviewer)
Comments and Suggestions for Authors
This manuscript presents a significant contribution to the field of pediatric neuro-oncology and transplant medicine through a large, nationwide, multicenter study focusing on Posterior Reversible Encephalopathy Syndrome (PRES). Its major strengths lie in the comprehensive characterization of the syndrome's prodromal phase, the proposal of a novel predictive index for early detection, and the critical analysis of long-term outcomes which challenges the classical notion of PRES as a fully "reversible" condition, prompting a timely discussion on terminology and future management strategies.
#1 The background section adds the pathophysiological mechanisms of PRES, especially the mechanism of endothelial damage and blood-brain barrier disruption in patients after chemotherapy/transplantation, and may cite recent studies on the mechanisms of PRES induced by drugs such as calmodulin neural phosphatase inhibitors, VCR, and immunosuppressants.
#2 The Methods section clarifies the diagnostic process for PRES: state whether all cases were confirmed by cranial MRI and reviewed by two independent neuroradiologists or neurologists.
#3 The discussion section delves into the relationship between PRES and survival. Is it due to intermediate factors such as treatment interruption, infection, and impaired organ function? It is recommended to discuss whether PRES is an independent prognostic factor or just a marker of disease severity.
#4 The discussion section needs to be more in-depth and provide important direction for subsequent research. This article focuses on the clinical features, risk factors, and prognosis of reversible posterior encephalopathy syndrome (PRES) presenting after childhood malignancy or hematopoietic stem cell transplantation, and systematically analyzes the syndrome's prodromal symptoms, laboratory abnormalities, and imaging manifestations and their impact on survival in a nationwide multicenter study with innovative PRES predictive indices and revised diagnostic terminology of the syndrome. The clinical prediction model constructed in this study is highly compatible with the idea of AI-assisted diagnosis, and the multidimensional clinical data it collects provides a valuable data base for the future development of a machine learning-based PRES early warning system, demonstrating the potential of transforming traditional clinical research into an intelligent medical tool. Indeed, recently, AI has made impressive progress in medical research and medical diagnostic treatments. For example, large language model (PMID: 41133425) and so on. In the discussion section, the authors need to first summarize the recent advances in AI and cite the above important literature as an introduction, and further re-examine the important inspirations of AI for this study, which will provide an important reference value for the subsequent studies.
Given these considerations, I highly recommend that authors revise their manuscript. Looking forward to receiving your revised version of the manuscript. I will review this manuscript again based on the revised version.
Author Response
Please see the attachment.

Reviewer 2 Report (New Reviewer)
Comments and Suggestions for Authors
Thank you for the opportunity to evaluate the authors' responses to the reviewers' questions. In my opinion, the authors have done a great job of improving the manuscript's content. They clearly and concisely stated their position on each issue and made appropriate adjustments, significantly enhancing the quality of the presented study. This study is of great interest to both pediatric oncologists and other medical specialists, as it addresses the complications of cancer progression and treatment, particularly acute leukemia. The occurrence of PRES significantly complicates the treatment of these children, as it delays necessary chemotherapy and contributes to increased rates of hospitalization in the ICU.
Author Response
Please see the attachment.

Reviewer 3 Report (New Reviewer)
Comments and Suggestions for Authors
This is a well researched study and a well written article describing the occurrence of posterior reversible encephalopathy syndrome among Polish youth treated for malignancies. The authors do a good job of describing their sample, and then perform statistical analyses to identify risk factors that impacted the likelihood of PRES and survival. Their predictive index and diagnostic criteria are helpful, although it is unclear what is gained by proposing a new name for the syndrome.
Round 2
Reviewer 1 Report (New Reviewer)
Comments and Suggestions for Authors
The authors have made significant revisions and efforts to improve their manuscript and have thoroughly resolved previous concerns. I recommend this manuscript for publication in its current form.
This manuscript is a resubmission of an earlier submission. The following is a list of the peer review reports and author responses from that submission.
Round 1
Reviewer 1 Report
Comments and Suggestions for Authors
This is a large study of pediatric cancer patients who had PRES; the literature is in need of large cohort studies using rigorous criterion for analysis. The author's report is compromised by the large volume of variables collected for analysis creating a high risk for Type I error; the authors need to justify their design and to take steps to support the notion that this statistical limitation was not encountered.
PRES unfortunately does not have a clear consensus definition and thus, is somewhat a diagnosis of exclusion; although the inclusion criterion was included, excluding other etiologies for the clinical presentation were not clearly stated. This is particularly relevant for CNS tumors where direct disease involvement could lead to some of the clinical manifestations.
Clinical symptoms and laboratory abnormalities were examined and implicated but there was no clear criterion on what constituted a variable that was related to PRES; many of the variables examined occurred days before the PRES was manifested thus challenging their significance. An explanation and justification of the inclusion of variables used is needed.
The authors found that PRES was linked to survival but provided no mechanism. It is not intuitively obvious why PRES would be associated with survival unless it altered the treatment plan of the patient.
The discussion section is long and unorganized; and does not clearly link past publications to this report in a consistent fashion. A substantial rewriting of this section is needed.
Comments on the Quality of English Language
There are minor type errors but the main issue is the organization of the text without a concise progression in thought that is easy to follow.
Author Response
Posterior reversible encephalopathy syndrome in children with malignancies or after hematopoietic cell transplantation: A Polish nationwide study
|
Response to Reviewer 1 Comments |
||
|
|
|
|
|
Thank you very much for taking the time to review this manuscript. Please find the detailed responses below and the corresponding revisions highlighted in the re-submitted files. |
||
|
|
|
|
|
|
||
|
Comments 1: This is a large study of pediatric cancer patients who had PRES; the literature is in need of large cohort studies using rigorous criterion for analysis. The author's report is compromised by the large volume of variables collected for analysis creating a high risk for Type I error; the authors need to justify their design and to take steps to support the notion that this statistical limitation was not encountered.
|
||
|
Response 1: We appreciate the reviewer's concern regarding the Type I error related to the number of variables analyzed. To address this, results of the comparisons between groups were additionally adjusted for multiple testing have been included (Table S1 in the Supplementary materials). After adjustment, the previously significant differences for hypertension before malignancy and hypertension during follow-up became non-significant, although they remained at the level of a statistical trend. Regarding the regression analyses, these were performed in two steps. The univariate analyses served as an exploratory step to identify variables potentially associated with PRES prevalence and to guide variable selection for the multivariate logistic regression models. This approach follows established modeling recommendations (e.g., Hosmer & Lemeshow, Applied Logistic Regression, 2nd ed.), where a liberal threshold (p<0.25) is used in univariate screening to ensure inclusion of potential confounders, thus the adjustment for multiple testing was not applied at this stage. The final multivariate models included only the most relevant predictors, thereby minimizing the risk of spurious associations due to multiple comparisons while ensuring model parsimony and interpretability. We added respective information in the text in section 3.2 and additional Table S1 in Supplementary materials.
Comments 2: PRES unfortunately does not have a clear consensus definition and thus, is somewhat a diagnosis of exclusion; although the inclusion criterion was included, excluding other etiologies for the clinical presentation were not clearly stated. This is particularly relevant for CNS tumors where direct disease involvement could lead to some of the clinical manifestations.
|
||
|
Response 2: PRES was diagnosed in patients with characteristic neuroimaging findings accompanied by a combination of specific clinical symptoms. In all patients included to the study group brain MRI was performed. Only patients with typical for PRES lesions and with no other findings which could explain the symptoms were included to the study. Diagnosis of PRES were made in a local pediatric oncology center by either oncologist and radiologist. Prior to final decision to include patient to the study, the diagnosis were confirmed by two researchers coordinating multicenter PRES project (TB and JS). Respective information was added in the text (section 2.2). |
||
|
Comments 3: Clinical symptoms and laboratory abnormalities were examined and implicated but there was no clear criterion on what constituted a variable that was related to PRES; many of the variables examined occurred days before the PRES was manifested thus challenging their significance. An explanation and justification of the inclusion of variables used is needed. |
||
|
Response 3: Thank you for that valuable comment. Previously published studies on PRES provided data on the type and frequency of symptoms. Until now no report emphasized a prodromal period. We included data on clinical symptoms and laboratory abnormalities. Further research is needed to confirm their significance in the pathogenesis and diagnosis of PRES. Based on our observations, we proposed the predictive index, which will be validated in a prospective study. We believe the predictive index may contribute to a faster PRES diagnosis thereby new possibilities for potential therapeutic interventions will be available. Respective information was added in the text (section 4.5) |
||
|
Comments 4: The authors found that PRES was linked to survival but provided no mechanism. It is not intuitively obvious why PRES would be associated with survival unless it altered the treatment plan of the patient. |
||
|
Response 4: The pathophysiology of reduced survival after PRES in children undergoing anticancer therapy or hematopoietic cell transplantation (HCT) is not fully understood, but it has been documented in both original studies (Zama, 2018, Eur J Haematol, Banerjee, 2018, J Pediatr Hematol Oncol; Gaziev, 2017, Biol Blood Marrow Transplant; Cordelli, 2021, Eur J Paediatr Neurol, Sommers, 2022) and a systematic review (Hun, 2020, Front Neurol). Although deaths directly related to PRES were reported (Zama, 2018), in most cases there are no simple connection; death is most often a result of progression or treatment toxicity. One of possible hypothesis assume that delaying therapy due to severe general condition during PRES may reduce treatment efficacy and increase the risk of progression or relapse, however there are no prospective studies confirming it. Respective information was added in the text (section 4.4).
Comments 5: The discussion section is long and unorganized; and does not clearly link past publications to this report in a consistent fashion. A substantial rewriting of this section is needed. |
||
|
Response 5: Thank you for that comment. Discussion was shortened. Chapter was divided into subparts: risk factors, clinical symptoms and laboratory abnormalities, early and late complications of PRES, survival, diagnostic criteria and predictive index, summary of discussion. To make it clear we added subtitles to each sections of the discussion. We also added additional explanations in sections 4.1 and 4.2, 4.4 and 4.5. |
||
|
4. Response to Comments on the Quality of English Language |
|
Point 1: The English could be improved to more clearly express the research. |
|
Response 1: The English was checked for style and grammar. |
|
5. Additional clarifications |
Our study is a part of nationwide reaserach project focused on PRES in children with malignacies or after HCT conducted by Polish Society of Pediatric Hematology and Oncology since 2022. Our society associated all Polish Pediatric Oncology and Hematology Centers and HCT units.
The PRES nationwide project in the near future assumes:
- further collection of data regarding new PRES episodes
- retrospective study concerned PRES in ALL (acute lymphoblastic leukemia) children in Poland between 2014-2025 with control group included all ALL patients reported in the ALL Registry
- retrospective study focused on correlation between radiological finding and clincal course of PRES
- prospective study testing and validaiting presented PRES predictive index
- prospective interventional study with protective administration of levetiracetam in children with PRES diagnosed prior to the seizures (main goal: to avoid status epilepticus, ICU admission and treatment delay)
Reviewer 2 Report
Comments and Suggestions for Authors
Brzeski and colleagues conducted a retrospective multicenter study on posterior reversible encephalopathy syndrome (PRES) in Poland. They proposed a PRES predictive index based on 120 pediatric cases. I commend the authors for completing this comprehensive nationwide study. However, the manuscript lacks several essential methodological details and requires further clarification and revision before it can be considered for publication.
Major
Line 130–140: Please cite prior literature when defining PRES. Including seminal references would strengthen the rationale for your diagnostic definition.
Line 141–144: Clarify the time frames for OS and DFS (from when to when). Additionally, ensure that OS and DFS are defined and applied consistently throughout the manuscript.
Line 151–: In multivariate analyses, variable selection should be performed before analysis based on biological and clinical rationale, not post-hoc univariate results. This is a major methodological concern and should be corrected.
Line 163: Please specify the total number of participating institutions to clarify the multicenter scope of the study.
Lines 114–124, 163–165: The control group included 318 patients, whereas 120 patients developed PRES. This incidence appears unexpectedly high. In the Discussion section, please discuss potential explanations for this frequency (e.g., diagnostic criteria, referral bias, or institutional differences).
Table 1: Radiographic findings would be informative for the reader. Please consider adding neuroimaging data such as lesion distribution or pattern (e.g., parieto-occipital, frontal, cerebellar) to the table.
Lines 204–212: Because laboratory findings (e.g., electrolyte disturbances, CRP elevation) may vary with disease type, please describe these abnormalities by underlying diagnosis or treatment phase if possible.
Table 2: Clarify when hypertension, seizures, and therapies occurred (e.g., before, during, or after PRES onset). The timing of these events is essential for interpreting causality.
Discussion: Please include a description of the time to onset of PRES, as this information is clinically relevant and not clearly presented.
The proposed PRES predictive index is potentially useful; however, the index should be validated after the above issues are addressed.
Lines 309–310: In the Discussion, clarify that older age appeared to increase the incidence of PRES according to your dataset, and discuss possible mechanisms or references supporting this trend.
Lines 325–328: If referenced data are part of your own cohort, please present them explicitly in the Results section rather than referring to external sources.
Minor
Figure 1: Spell out “HCT” in the figure legend and axis labels for clarity.
Lines 313–315: If sickle cell disease (SCD) cases were not included in your cohort, these sentences are not applicable and should be deleted or revised for relevance.
Abbreviations: Please abbreviate recurring terms only once (e.g., ALL) and ensure consistent usage throughout the text.
Author Response
Posterior reversible encephalopathy syndrome in children with malignancies or after hematopoietic cell transplantation: A Polish nationwide study
|
Response to Reviewer 2 Comments |
||
|
|
|
|
|
Thank you very much for taking the time to review this manuscript. Please find the detailed responses below and the corresponding revisions highlighted in the re-submitted files. |
||
|
|
||
|
Comments 1: Line 130–140: Please cite prior literature when defining PRES. Including seminal references would strengthen the rationale for your diagnostic definition.
|
||
|
Response 1: Thank you for the comment. We added respective citations (section 2.4).
|
||
|
Comments 2: Line 141–144: Clarify the time frames for OS and DFS (from when to when). Additionally, ensure that OS and DFS are defined and applied consistently throughout the manuscript.
|
||
|
Response 2: Thank you for this valuable comment. We clarified in the Methods section 2.5 that overall survival (OS) was calculated from the date of cancer diagnosis to the date of death from any cause, last follow-up, or study completion, and disease-free survival (DFS) from the date of diagnosis to the date of recurrence or death (whichever occurred first), last follow-up, or study completion. |
||
|
Comments 3: Line 151–: In multivariate analyses, variable selection should be performed before analysis based on biological and clinical rationale, not post-hoc univariate results. This is a major methodological concern and should be corrected. |
||
|
Response 3: Thank you for this important comment. We agree that variable selection in multivariate models should be guided by clinical and biological rationale. In our analysis, variables were first selected for the study based on their clinical relevance and then screened in univariate logistic regression, with those showing p<0.25 included in the multivariate model. This approach follows the recommendations of Hosmer and Lemeshow (Applied Logistic Regression, 3rd edition, 2013), who suggest this threshold as an appropriate criterion to avoid excluding potentially important variables. The final model was derived using stepwise selection among the pre-screened variables. This approach is widely used in medical research, providing objectivity while allowing the identification of potentially novel associations that may not be evident. We added respective position in References – [34]. |
||
|
Comments 4: Line 163: Please specify the total number of participating institutions to clarify the multicenter scope of the study.
|
||
|
Response 4: Thank you for pointing out at this issue. Total number of participating institutions was 17, we added it to the text (section 3.1)
Comments 5: Lines 114–124, 163–165: The control group included 318 patients, whereas 120 patients developed PRES. This incidence appears unexpectedly high. In the Discussion section, please discuss potential explanations for this frequency (e.g., diagnostic criteria, referral bias, or institutional differences). |
||
|
Response 5: Thank you for the comment. All children with PRES occurred between 2014-2022 during anticancer treatment or undergoing HCT were included to the study group. Due to the diverse distribution of diseases and the multiple of analyzed parameters, selecting a control group was a particular issue. For each patient with PRES, 2-3 control patients with the same disease or undergoing HCT were selected locally. We are aware of the risk of potential subjectivity and referral bias therefore verification whether the groups do not differ significantly have been done. There were no significant differences in types of disease, gender, percentage of HCT patients or CNS involvement. Although the control group selection could be consider as a limitation of the study, as mentioned in the discussion section, nevertheless our aim was to analyze all pediatric PRES cases associated with anticancer treatment or HCT in Poland over analyzed period. Assuming average number of 1150 children diagnosed for malignancy each year in Poland, including 230 children with ALL we estimated frequency of PRES occurring in Poland for 1 episode per 85 children with malignancy (1,2%) and 1 episode per episode per every 30 children with ALL (3,0%). We added respective information to the text (section 4). We also added appropriate citations to the references confirming the number of newly diagnosed children with malignancy each year in Poland – [35].
Comments 6: Table 1: Radiographic findings would be informative for the reader. Please consider adding neuroimaging data such as lesion distribution or pattern (e.g., parieto-occipital, frontal, cerebellar) to the table. |
||
|
Response 6: Thank you for the comment. In Table 1 we focused on comparing the study group with the control group using parameters available for both groups. Detailed analysis of the radiological image was not the aim of our study. Currently a detailed analysis of the correlation between the radiological image and the clinical course of PRES, as a continuation of this project is ongoing. The analysis will include not only the PRES imaging pattern proposed by Bartynski et al. but also detailed anatomical locations, contrast enhancement, diffusion restrictions. Please find the description of our PRES nationwide project in section "5.Additional clarifications".
|
||
|
Comments 7: Lines 204–212: Because laboratory findings (e.g., electrolyte disturbances, CRP elevation) may vary with disease type, please describe these abnormalities by underlying diagnosis or treatment phase if possible. |
||
|
Response 7: Thank you for the comment. Please find the Table S2 in the Supplementary materials which provide more data regarding the types of disease in specific laboratory findings (CRP elevation, electrolyte disturbances, hypertransaminasemia). Median time from the diagnosis of de novo disease to onset of PRES in CRP elevated group: 1.4 months. Median time from the diagnosis of de novo disease to onset of PRES in electrolyte disturbances group: 1.4 months. Median time from the diagnosis of de novo disease to onset of PRES in hypertransaminasemia group: 1.4 months. Due to median time from the diagnosis of de novo disease to onset of PRES in specific laboratory findings is very close to median time from the diagnosis of de novo disease to onset of PRES in all study group patients, we did not present it in the manuscript. |
||
|
Comments 8: Table 2: Clarify when hypertension, seizures, and therapies occurred (e.g., before, during, or after PRES onset). The timing of these events is essential for interpreting causality. |
||
|
Response 8: Table 2 includes the results of univariate and multivariate analyses for the occurrence of PRES. During data collection process co-investigators received the same question to both study group (PRES) and control group (non-PRES) to enable comparison: "hypertension during therapy or after HCT" and "seizures during therapy or after HCT". In all patients from the study group hypertension and seizures occurred before diagnosis of PRES (during prodromal period when clinical symptoms were observed but before neuroimaging confirmation). We added adequate comment under the table 2. |
||
|
Comments 9: Discussion: Please include a description of the time to onset of PRES, as this information is clinically relevant and not clearly presented. |
|
Response 9: Thank you for pointing this. We added adequate sentence in discussion in section 4.2. Majority of PRES episodes in ALL occurred during induction chemotherapy. In Banerjee et al. study 28/29 children with PRES was diagnosed during protocol I. Similar conclusion was reported by Zama et al. where 65/67 PRES cases occured during first 30 days of ALL chemotherapy; these authors emphasize that PRES occurred later in children undergoing HCT than after chemotherapy. Our results also correspond with those presented by Anastasopoulou et al. where 28/52 PRES episodes observed during induction therapy. |
|
Comments 10: The proposed PRES predictive index is potentially useful; however, the index should be validated after the above issues are addressed. |
|
Response 10: Thank you for that valuable comment. Previously published studies on PRES provided data on the type and frequency of symptoms. Until now no report emphasized a prodromal period. Our results regarding clinical symptoms and laboratory abnormalities are observational data. The authors cannot confirm their significance in the pathogenesis or diagnosing PRES. Further research is planned. Based on our observations, we proposed the predictive index, which will be validated in a prospective study. We believe the predictive index may contribute to a faster PRES diagnosis thereby new possibilities for potential therapeutic interventions will be available.
|
|
Comments 11: Lines 309–310: In the Discussion, clarify that older age appeared to increase the incidence of PRES according to your dataset, and discuss possible mechanisms or references supporting this trend. |
|
Response 11: Thank you for the comment. According to our results and those presented by others (Thavamani, 2020, Pediatr Neurol., and Anastasopoulou, 2019, Pediatr Blood Cancer) older age increase risk of PRES. There is no leading theory which explains that finding. In our opinion some hypothesis e.g. increased dose of potentially neurotoxic drugs associated with higher body mass or protective influence of less mature CNS to autoregulation of cerebral blood flow could be considered, but there is no data providing evidences. In contrast to this studies another authors (Sommers, 2022, J Pediatr Hematol Oncol) reported no connection between older age and PRES. Impact of older age on PRES remains unclear. Further studies are needed. We added the comment in discussion; section 4.1.
|
|
Comments 12: Lines 325–328: If referenced data are part of your own cohort, please present them explicitly in the Results section rather than referring to external sources. |
|
Response 12: Thank you for that comment. We corrected this issue in revised manuscript.
|
Minor
|
Comments 13: Figure 1: Spell out “HCT” in the figure legend and axis labels for clarity.
|
|
Response 13: Thank you for that comment. Figure 1 was enlarged and made clearer.
|
|
Comments 14: Lines 313–315: If sickle cell disease (SCD) cases were not included in your cohort, these sentences are not applicable and should be deleted or revised for relevance.
|
|
Response 14: Thank you for that comment. We deleted these sentences in revised manuscript.
|
|
Comments 15: Abbreviations: Please abbreviate recurring terms only once (e.g., ALL) and ensure consistent usage throughout the text. |
|
Response 15: Thank you for that comment. We corrected this issue in revised manuscript.
|
|
4. Response to Comments on the Quality of English Language |
|
Point 1: The English is fine and does not require any improvement. |
|
Response 1: Thank you. |
|
5. Additional clarifications |
Our study is a part of nationwide reaserach project focused on PRES in children with malignacies or after HCT conducted by Polish Society of Pediatric Hematology and Oncology since 2022. Our society associated all Polish Pediatric Oncology and Hematology Centers and HCT units.
The PRES nationwide project in the near future assumes:
- further collection of data regarding new PRES episodes
- retrospective study concerned PRES in ALL (acute lymphoblastic leukemia) children in Poland between 2014-2025 with control group included all ALL patients reported in the ALL Registry
- retrospective study focused on correlation between radiological finding and clincal course of PRES
- prospective study testing and validaiting presented PRES predictive index
- prospective interventional study with protective administration of levetiracetam in children with PRES diagnosed prior to the seizures (main goal: to avoid status epilepticus, ICU admission and treatment delay)